# *HLA-DQB1*06* and Select Neighboring HLA Variants Predict Chlamydia Reinfection Risk

**DOI:** 10.3390/ijms242115803

**Published:** 2023-10-31

**Authors:** Kanupriya Gupta, Howard W. Wiener, Hemant K. Tiwari, William M. Geisler

**Affiliations:** 1Department of Medicine, Division of Infectious Diseases, University of Alabama at Birmingham, Birmingham, AL 35294, USA; kanupriyagupta@uabmc.edu; 2Department of Epidemiology, University of Alabama at Birmingham, Birmingham, AL 35294, USA; hwiener@uab.edu; 3Department of Biostatistics, University of Alabama at Birmingham, Birmingham, AL 35294, USA; htiwari@uab.edu

**Keywords:** *Chlamydia trachomatis*, HLA, reinfection

## Abstract

Associations of HLA class II alleles with genital chlamydial infection outcomes have been reported, especially *HLA DQB1*06*. However, the potential role of *DQB1*06* in influencing reinfection risk has still not been established. The purpose of this study was to determine whether the association of *DQB1*06* with chlamydia reinfection was impacted by any other nearby HLA class II variants that were also associated with reinfection. We used next-generation sequencing to map HLA class II variants spanning the *HLA-DQ* and -*DR* loci. *DQB1*06* as well as *DQB1*04* were confirmed as significant predictors of chlamydia reinfection, when controlling for age and percent African ancestry. SKAT analysis revealed one region each in *DRB1*, *DRB5*, *DQA2*, and three intergenic regions that had variants associated with reinfection. Further analyses of these variants revealed that rs112651494 within *DRB5* and an intergenic SNP rs617058 in *DRB1*:*DQA1* were significantly associated with reinfection, but this did not impact the significance of the association of *DQB1*06* or *DQB1*04* with reinfection.

## 1. Introduction

Chlamydia remains one of the most prevalent bacterial sexually transmitted infections worldwide [1], including in the U.S. where over 1.6 million cases are reported annually [2]. Women are disproportionally affected by chlamydia due to reproductive and perinatal morbidity, and women younger than age 25 years account for the majority of chlamydia cases in the U.S. [2]. Racial disparities also exist in reported chlamydia rates in the U.S., with a ~6 fold higher rate in those who self-report as African American versus Caucasian [2]. Socioeconomic factors and recent sexual behaviors are not consistent predictors of chlamydia risk [3], and there may be other factors in some individuals that contribute to the burden of chlamydia in African American women; immunogenetic factors likely play a role. Chlamydia reinfection is very common, occurring in up to 20% of infected individuals within one year after treatment [4], and it increases the risk for chlamydia complications and continued chlamydia transmission. This finding that reinfection occurs commonly within a short time period supports the notion that chlamydia immunity is short-lived in some. Elucidating immunogenetic factors contributing to protection against or risk for reinfection could help to identify potential biomarkers of chlamydia risk and be useful for chlamydia vaccine development.

We previously showed that *Chlamydia trachomatis* (CT)-specific systemic CD4^+^ IFN-γ production is an immune correlate of protection against chlamydia reinfection [5], which is consistent with murine chlamydia model findings [6]. Given that HLA class II coded molecules process exogenous peptides for presentation to CD4^+^ helper T cells [7], we have also investigated the association of HLA class II alleles with reinfection. We previously found in two unique cohorts that *HLA-DQB1*06* was associated with chlamydia reinfection risk: a multicenter cohort of North American adolescents at increased risk for sexually transmitted infections (STIs) [8] and a cohort of women who self-reported as African American race seen at a sexual health clinic in Birmingham, AL [9]; this allele has also been associated with pelvic inflammatory disease (PID) [10] and infertility [11]. We then reported data from a Bayesian model in the same Birmingham cohort that modeled the impact of *DQB1*06* and CT-specific systemic CD4^+^ IFN-γ on reinfection probability that showed that the highest probability of reinfection (62%) was in women with *DQB1*06* but without a CT-specific CD4^+^ IFN-γ response [12].

Despite the consistent association of *HLA-DQB1*06* with chlamydia reinfection risk, these association studies do not establish *DQB1*06* as a genetic variant driving the chlamydia outcomes because of possible other neighboring HLA class II variants that could impact association of *DQB1*06* with reinfection; these other variants could be in linkage disequilibrium (LD) with *DQB1*06*, which has been well described between DQ and DR loci [13]. Before pursuing mechanistic studies on *HLA-DQB1*06* and chlamydia reinfection risk and its consideration as a biomarker for reinfection, a more comprehensive evaluation of other HLA class II variants that may be impacting the *DQB1*06* reinfection association is needed. The purpose of our study was to determine whether HLA class II variants neighboring *HLA-DQB1*06* impacted its predicted reinfection risk in our Birmingham cohort. We sequenced the MHC class II region spanning *HLA-DQ* and -*DR* and used a fine-mapping approach to assess significance of nearby variants. Because our earlier study in the Birmingham cohort was based on self-reported race, we also incorporated percent African ancestry in our HLA class II variants and reinfection risk analyses since HLA allele distribution patterns have been shown to be associated with specific populations with shared demographic history [14].

## 2. Results

### 2.1. Participant Characteristics

This study included 165 chlamydia-infected African American women from a cohort whom we previously evaluated for association of specific *HLA-DQB1* alleles with chlamydia reinfection at 3 and 6 month follow-up visits after treatment [9]. Baseline participant characteristics were previously described [9]. The median age was 22, 51% were asymptomatic, 26% had bacterial vaginosis, and 51% reported prior chlamydia. Chlamydia reinfection occurred in 38 (20%) participants [9]. In the current study, percent African ancestry for each participant was determined using whole genome genotypes data (discussed in Section 4.3). Table 1 shows the distribution of age, percent African ancestry, and *DQB1* alleles in the study cohort.

### 2.2. DQB1*06 and DQB1*04 Were Predictors of Chlamydia Reinfection

Figure 1 describes the analysis plan for association testing of chlamydia reinfection status with *HLA-DQB1* alleles and sequencing data from the chromosome 6 region *HLA-DRA* through *HLA-DQB1*. In our previous study, associations of *HLA-DQB1* alleles with reinfection were analyzed without controlling for African ancestry. In the current study, we modeled percent African ancestry as a covariate (discussed in Section 4.3). Logistic regression models for chlamydia reinfection were examined that included age, proportion of African ancestry, and one of the *DQB1* genotypes. The allele of *DQB1*06* was significant, and that of *DQB1*04* was marginally significant with Bonferroni correction (Table 2). When we looked at a model that included all of the DQB1 genotypes, both *DQB1*04* and *DQB1*06* alleles were significant (see Table A1). Furthermore, significance for *DQB1*04* only remained when both *DQB1*04* and *DQB1*06* were included the model (Table 3), whereas *DQB1*06* was significantly associated in the absence of DQB1*04. The significance level of both *DQB1*04* (*p* = 0.004) and *DQB1*06* (*p* = 0.002) improved in the joint modeling. A similar pattern was seen in models that did not include proportion of African ancestry. Among women with a *DQB1*06* allele, most had a *DQB1*06:02* allele (63 of 81 [77.78%]), whereas frequencies of other *06 alleles were lower (**06:03*, 13.58%; **06:04*, 11.11%; and **06:09*, 8.64%). (Nine women had two different alleles, so these proportions add to more than 100%). Among those with *a DQB1*04* allele, the majority had a *DQB1*04:02* allele (13 of 19 [68.42%]) and **04:01* was less common (31.58%). Due to the small sample sizes of alleles at the four-digit specificity level, they were not included in the logistic regression model and further analyses were conducted using **06* and **04* at the two-digit specificity.

### 2.3. Regions in DRB1, DRB5, and DQA2 and also Intergenic Regions Were Associated with Chlamydia Reinfection

In the current study, DNA from 165 participants was used to capture the chromosome 6 region *HLA-DRA* through *HLA-DQB1* using custom oligos (Agilent) and subjecting to paired end 150 bp sequencing on the Illumina NextSeq500 to determine whether additional HLA variants predicted reinfection risk independent of *DQB1*06* and *DQB1*04*. Sequence Kernel Association Test (SKAT) [15] was used on moving windows (of size 5, 10, 20, or 50 variants) through the sequenced region to test for association between windows and reinfection. Age, African ancestry, and presence of both *HLA-DQB1*06* and *HLA-DQB1*04* were included as covariates. SKAT results revealed one region each in *DRB1*, *DRB5*, and *DQA2*, as well as three intergenic regions with variants that were significantly associated with reinfection with *p* < 0.05 (Table 4). Note that in Table 4, the window of five variants had the lowest *p*-values for these regions compared to other window sizes. Table A2 provides the names of variants, positions, alleles, minor allele frequency (MAF), and annotations of 30 variants within all six windows. Out of 13 variants (identified in windows of variants associated with reinfection, Table 4), only 7 were independent variants, implying a significance level of ~0.007 (=0.05/7) using Bonferroni correction.

### 2.4. Variants Associated with Reinfection

We further performed single variant analysis from the window of five variants within the six HLA regions that were associated with reinfection on SKAT analyses. We analyzed the variants with MAF ≥ 0.05 due to small sample size using a logistic regression model to detect variant association with chlamydia reinfection including *DQB1*04* and **06*, age, and percent African ancestry as covariates. Variants rs112651494 (within *HLA-DRB5*) and rs617058 (within *HLA-DRB1:HLA-DQA1*) were marginally significant (Table A3). Furthermore, we included both marginally significant variants with other covariates (*DQB1*04* and **06*, age, and percent African ancestry) in a logistic regression model to observe the joint effect of the putative variants. In this new joint model, rs112651494 (*p* = 0.009) and rs617058 (*p* = 0.018) had stronger association with chlamydia reinfection than previously observed (Table 5); however, both rs112651494 and rs617058 were only marginally significant after multiple testing correction (see Section 4.6).

## 3. Discussion

Determining HLA variants that strongly predict reinfection risk can facilitate analyses of the immune mechanisms and effector responses mediated by those variants of interest and may aid in identifying CT epitopes among vaccine antigen candidates. Having shown a consistent association of *DQB1*06* with reinfection risk in two different cohorts [8,9], this study was focused on determining whether it is *DQB1*06* and/or other nearby HLA variants in LD that strongly predict reinfection risk through applying next-generation sequencing to fully interrogate the HLA class II region spanning the *HLA-DQ* and -DR, which is important given strong LD between *DQ* and *DR* loci [13,16].

Our initial evaluation involved controlling for percent African ancestry in the analysis of *DQB1* alleles associated with reinfection, unlike in our previous study [9] and we now had access to variants data acquired through NGS that could aid in racial/ethnic stratification. It is important to control for African ancestry in analyses of genetic markers associated with chlamydia outcomes because higher chlamydia rates have been found to be associated with reported Black race in young females [2,17]. Using a logistic regression model that included chlamydia reinfection as an outcome and one of the *HLA-DQB1* alleles as a predictor, we confirmed the association of *DQB1*06* with reinfection risk and additionally found *DQB1*04* to be associated with reinfection. This association remained significant for *DQB1***04* only when both *DQB1*04* and *DQB1*06* were included in the model (while *DQB1*06* was associated with reinfection independent of *DQB1*04*); the greater significance of the association for both variants in the joint model corroborates that both *DQB1* alleles are likely important predictors of chlamydia reinfection. A recent study looking at the effects of *HLA-DRB1-DQB1* alleles/haplotypes’ effect on chlamydial infection outcome found only *DQB1**05:03:01 G was associated with chlamydia clearance and persistence events. They did not find *DQB1***06* or *DQB1***04* alleles associated with these *chlamydia* outcomes, but this could be due to differences in genetic background (South American vs. African American) [18]. Genotyping of commercial sex workers whose data previously demonstrated a correlation between Chsp60 antibody and risk of *chlamydia* PID identified *DRB1, DQA1*, and *DQB1* alleles were predominant in this population; *HLA DQA1*0401/DQB1*0402* heterodimer was associated with increased Chsp60 antibodies [19]. Pedraza et al. found 16 *HLA-DRB1* alleles having significant effect on chlamydial infection outcome [18]. *HLA-DRB1* allele homozygous women were associated with events having a lower probability of clearance and/or early occurrence of persistence. They also found 142 *DRB1-DQB1* haplotypes that had statistically significant values when analyzing the effect of each haplotype on chlamydia outcome including *HLA-DQB1*06* and **04* when configured as haplotypes with DRB1 allele [18].

To our knowledge, this is the first study in the chlamydia field that went beyond HLA genotyping in order to identify specific variants in the HLA class II alleles associated with chlamydia reinfection. Our comprehensive sequencing followed by SKAT analysis of the *HLA-DRA* through *HLA-DQB1* region revealed six additional HLA class II regions with windows of variants that were significantly associated with reinfection independent of *DQB1*06* and *DQB1*04*, with one region each in *DRB1*, *DRB5*, and *DQA2*, as well as three intergenic regions. Further, we proceeded to perform single variant analysis within these six HLA regions and found that rs112651494 within *DRB5* and an intergenic SNP rs617058 in *DRB1*:*DQA1* were significantly associated with reinfection. Since rs112651494 and rs617058 are found in the intronic and the intergenic regions, respectively, and do not code for any amino acids, it would be interesting to elucidate whether these variants have any regulatory effects at the transcriptional level that can affect the binding of chlamydial antigens to the HLA class II molecules and, in turn, modify the activation of CD4^+^ T cells which are important for protection against chlamydia reinfection. In the literature, none of these variants have been found to be associated with any infection risks. Interestingly, when both rs112651494 and rs617058 were included in a logistic regression model to observe the joint effect of these two putative variants with the other covariates including *DQB1*04* and **06*, they remained significant, suggesting a lack of LD with these *DQB1* alleles.

A limitation of our study was the small sample size, which not only limited analyses to two-digit specificity of *HLA-DQB1*04* and **06*, but also could have impacted the significance of our results; thus, the findings need to be replicated in a larger population. Our study was limited in that it focused on women self-reporting as Black race, so it is unknown if findings can be generalized to those of other races/ethnicities; the frequency of *DQB1*04* (11.5%) and *DQB1*06* (49.1%) were higher in our cohort compared to that of U.S. African American women in the Allele Frequency Net Database (AFND), which are 7% and 29.5%, respectively [20]. Our study restricted analyses to HLA class II alleles and variants (identified through sequencing) because our earlier studies showed the association of *DQB1*06* with reinfection and CD4^+^ IFN-γ protecting against reinfection [5]. We did not evaluate HLA class I alleles or other variants in this study, in part because murine and human studies in general have not shown convincing evidence that HLA class I alleles, other immunogenetic variants, or CD8^+^ cells are associated with chlamydia outcomes, but this is an area that deserves further study. We also did not include CD4^+^ IFN-γ in our models because we did not have this data for all of the subjects analyzed in this study but this is a future direction of the research.

Our study verified that *DQB1*06* is a key putative contributor to reinfection risk along with *DQB1*04,* and both rs112651494 and rs617058 variants are also likely putative contributors, which supports our hypothesis that immunogenetic factors contribute to reinfection risk. Populations with higher frequency of these variants could have higher reinfection risk. One of the implications of this study could be that *DQB1*04* and **06* along with rs112651494 and rs617058 variants could be used as biomarkers for chlamydia reinfection risk and used to guide timing and frequency of repeat chlamydia testing after chlamydia treatment. Identifying additional genetics variants contributing to reinfection risk that we did not evaluate in our study, such as HLA class I alleles or cytokine gene, could also be used with the variates we identified to create genetic profiles to identify those at highest risk for reinfection and, thus, could be used to inform chlamydia testing practices. Knowledge of these risk variants and others may also be important in chlamydia vaccine development research, in terms of chlamydia antigen selection for a vaccine as well as understanding mechanisms of the human immune response to chlamydia antigens. Knowledge of specific HLA gene variants influencing reinfection risk will also inform T-cell epitope studies on vaccine candidates. This approach will also then facilitate biological validation in the context of adaptive immunity with analyses of the relationship of these HLA variants with CT-specific immune responses and mechanisms. Future studies will also address underlying functional mechanisms and immune responses through which these genetic variants increase chlamydia reinfection risk.

## 4. Materials and Methods

### 4.1. Study Design and Sample Collection

This study included 165 African American women (with samples available for HLA next-generation sequencing) from a study cohort subgroup of 185 women described in our previous work on associations of *HLA-DQB1* alleles with reinfection [9]. Briefly, these women were ≥16 years of age and presented to a sexual health clinic for treatment of a positive screening CT nucleic acid amplification test (NAAT), at which time they were enrolled (after providing written consent), treated with azithromycin 1 g single dose as directly observed therapy, and were tested for chlamydia reinfection using NAAT at 3 and 6 month follow-up visits; those with a positive CT NAAT at either follow-up visit were classified as reinfected and treated with the same azithromycin regimen as the enrolment visit. Blood was collected at each visit for immunologic and genetic studies.

### 4.2. HLA Capture and Sequencing

*HLA-DQB1* genotyping data used for this study had been previously determined via PCR and Sanger sequencing methods as previously described [9]. For this study, genomic DNA was then used to capture the chromosome 6 region *HLA-DRA* through *HLA-DQB1* using custom capture oligos designed using the Agilent SureDesign web tool to the interval containing *HLA-DRA* 6: 32,420,000 through *HLA-DQB1* 6: 32,700,000 using the GRCh38 (hg38) version of the human genome. DNA libraries were produced using the Agilent XT HS library kit as per the manufacturer’s instructions (Agilent, Santa Clara, CA, USA) prior to capture. Eighteen individually indexed libraries were combined for capture following the manufacturer’s recommendations (Agilent, Santa Clara, CA, USA). The captured libraries were quantitated via qPCR (Roche, Indianapolis, IN, USA), combined at equimolar amounts, and run on the Illumina NextSeq500 (Illumina, San Diego, CA, USA) with paired end 150 bp reads using standard techniques. Sequencing data was processed using GATK version 4.1.9. Only variants that passed the final VQSR quality control step were retained.

### 4.3. Percent Ancestry Estimation

We used the ADMIXTURE software program (version 1.3) to estimate ancestry in a model-based manner using autosomal SNP genotype data in unrelated case–control samples [21]. We assumed two ancestral populations (European and African) to estimate the proportion of ancestries in our samples. Since ADMIXTURE does not consider LD while estimating ancestry, we used only 55,061 tag variants with LD of r^2^ < 0.05.

### 4.4. Analysis of HLA-DQB1 Alleles and Reinfection

Logistic regression was used to examine the effects of genetic and non-genetic covariates on reinfection risk. An initial model included age and African ancestry in addition to the presence of *DQB1* alleles **02*, **03*, **04*, **05*, and **06*. We also analyzed the joint effect of all *DQB1* alleles including age and percent African ancestry in the model. Other non-genetic covariates were not included in the model since they were not associated with reinfection risk on univariate analysis.

### 4.5. Sequence Kernel Association Test (SKAT) Analysis

The null model used for the SKAT analysis included covariates: age, proportion of African ancestry, and presence of alleles *DQB1*06* and *DQB1*04*. HLA class II sequencing data were included in the SKAT analysis in windows of consecutive variants. Window sizes used for the analyses were 50, 20, 10, and 5 variants. Beginning at the start of the sequenced region, SKAT analysis was performed for the first set of consecutive variants of the window size. Then, the first variant was removed, and the next available variant added to the end of the window. This was continued until the end of the sequenced region was reached. For each window, two analyses were performed: SKATBinary_Robust analysis (*p*-value was reported as *p*(Rob)) [22] and SKAT_CommonRare analysis (*p*-value was reported as *p*(CR)) [23], to allow for windows that contained more common variants.

### 4.6. Analyses of Variants in HLA Class II Regions

Logistic regression was used to examine the effects of variants associated with chlamydia reinfection in the HLA regions with the smallest window size of variants associated with reinfection in SKAT analyses. All models included age and African ancestry as covariates. In addition, for multiple testing correction, we used Li and Ji method to determine effective/independent number of variants since several variants were in high linkage disequilibrium [24].

## Figures and Tables

**Figure 1 ijms-24-15803-f001:**
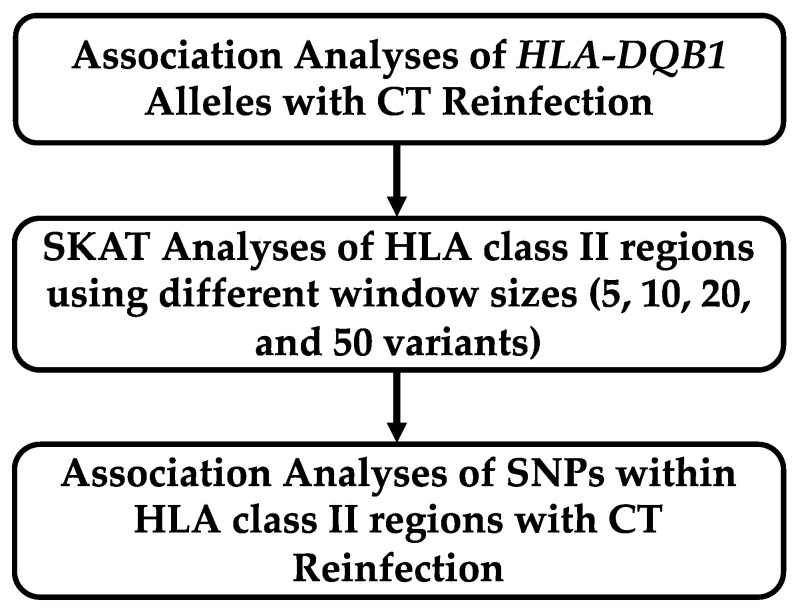
Analysis plan for association testing of chlamydia reinfection with HLA class II region.

**Table 1 ijms-24-15803-t001:** Distribution of the age, percent African ancestry, and *DQB1* alleles in the study cohort, stratified by chlamydia reinfection status.

Variable	No Reinfection (*n* = 128)	Reinfection (*n* = 37)	Total (*n* = 165)
Age, Mean (SD)	23.19 (4.52)	22.08 (3.81)	22.94 (4.39)
Percent African ancestry, Mean (SD)	0.75 (0.16)	0.73 (0.15)	0.75 (0.16)
^1^ *DQB1*02*, *n* (%)	44 (34.4)	8 (21.6)	52 (31.5)
^1^ *DQB1*03*, *n* (%)	51 (39.8)	11 (29.7)	62 (37.6)
^1^ *DQB1*04*, *n* (%)	12 (9.4)	7 (18.9)	19 (11.5)
^1^ *DQB1*05*, *n* (%)	53 (41.4)	13 (35.1)	66 (40.0)
^1^ *DQB1*06*, *n* (%)	56 (43.8)	25 (67.6)	81 (49.1)

^1^ Subjects carrying the specified allele.

**Table 2 ijms-24-15803-t002:** Association analyses of *HLA-DQB1* alleles ^1^ associated with chlamydia reinfection using a logistic regression model ^2^.

Variable	OR (95% CI)	*p*-Value
*DQB1*02*	0.52 (0.22–1.25)	0.145
*DQB1*03*	0.63 (0.28–1.40)	0.255
*DQB1*04*	2.97 (1.01–8.74)	0.048
*DQB1*05*	0.73 (0.34–1.58)	0.428
*DQB1*06*	2.66 (1.22–5.78)	0.014

^1^ *HLA-DQB1* alleles were determined by genotyping. ^2^ The logistic regression model included chlamydia reinfection as an outcome and one of the *HLA-DQB1* alleles as a predictor. All models included age and percent African ancestry as covariates.

**Table 3 ijms-24-15803-t003:** Association analyses of joint modeling of *DQB1*04* and *DQB*06* ^1^ with chlamydia reinfection using the logistic regression model ^2^.

Variable	OR (95% CI)	*p*-Value
Age	0.92 (0.83–1.02)	0.099
Percent African ancestry	0.15 (0.01–1.85)	0.139
*DQB1*04*	5.80 (1.71–19.74)	0.005
*DQB1*06*	3.94 (1.62–9.59)	0.002

^1^ Note that only *DQB1*04* (*p* = 0.013) and *DQB1*06* (*p* = 0.028) were significant with *p* < 0.05 in the joint modeling of all *DQB1* alleles (Table A1). ^2^ Model: chlamydia reinfection = age + percent African ancestry + *DQB*04* + *DQB*06.*

**Table 4 ijms-24-15803-t004:** HLA class II regions that had one or more windows of variants with a significant association (*p* < 0.05) with chlamydia reinfection using SKAT analysis ^1^.

Genes/Genomic Region	End Position ^2^	Number of Variants per Window	Start	Length	*p* (Rob) ^3^	*p* (CR) ^4^
*BTNL2*:*MIR5004*	32,430,971	5	32,429,118	1853	0.011	0.009
		10	32,428,330	2641	0.039	0.076
		20	32,427,550	3421	0.024	0.029
		50	32,426,255	4716	0.078	0.047
*HLA-DRB5*	32,528,218	5	32,528,189	29	0.024	0.005
		10	32,528,178	40	0.024	0.005
		20	32,528,113	105	0.024	0.009
		50	32,526,275	1943	0.024	0.009
*HLA-DRB5*:*HLA-DRB6*	32,541,699	5	32,541,683	16	0.038	0.010
		10	32,541,640	59	0.049	0.010
		20	32,541,543	156	0.096	0.087
		50	32,541,186	513	0.101	0.097
*HLA-DRB1*	32,580,623	5	32,580,536	87	0.013	0.009
		10	32,580,509	114	0.026	0.087
		20	32,580,180	443	0.066	0.173
		50	32,579,186	1437	0.172	0.373
*HLA-DRB1*:*HLA-DQA1*	32,606,702	5	32,606,320	382	0.011	0.005
		10	32,606,237	465	0.013	0.007
		20	32,606,100	602	0.098	0.223
		50	32,605,511	1191	0.148	0.318
*HLA-DQA2*	32,741,530	5	32,741,474	56	0.010	0.032
		10	32,741,412	118	0.011	0.036
		20	32,741,263	267	0.020	0.030
		50	32,739,501	2029	0.054	0.048

^1^ SKAT is testing the joint effect of the variants in a given window after taking the null model (age, percent African ancestry, *DQB1* alleles **04* and **06*) into account. ^2^ Each entry in the table shows windows of four different sizes that all end with the same SNP variant, so each set is identified using the common position of the ending SNP. ^3^ *p* (Rob) is the *p*-value for the robust test of the set of variants in the window. ^4^ *p* (CR) is the *p*-value for the common/rare test of the set of variants in the window.

**Table 5 ijms-24-15803-t005:** Association analysis of variants within HLA class II regions with chlamydia reinfection controlling for age, percent African ancestry, *DQB*04*, and *DQB*06* ^1^.

Variable ^1^	OR (95% CI)	*p*-Value
Age	0.925 (0.824–1.039)	0.190
Percent African ancestry	0.064 (0.002–1.739)	0.103
*DQB1*04*	12.156 (1.924–76.808)	0.008
*DQB1*06*	8.572 (2.176–33.774)	0.002
rs112651494	2.475 (1.254–4.883)	0.009
rs617058	2.811 (1.194–6.614)	0.018

^1^ Model: chlamydia-reinfection = age + percent African ancestry + *DQB*04* + *DQB*06* + rs112651494 + rs617058.

## Data Availability

Data used in the manuscript and analytic code will be made available upon request.

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
