# Peer review of "HLA-DQB1*06 and Select Neighboring HLA Variants Predict Chlamydia Reinfection Risk"

_ijms, 2023, doi:10.3390/ijms242115803_

Round 1

Reviewer 1 Report

Comments and Suggestions for Authors

Thanks for the opportunity to review the manuscript by Kanupriya Gupta and collaborators. In this study, the authors focused on determining whether DQB1*06 and other nearby HLA variants in LD strongly predict reinfection risk by applying NGS to thoroughly interrogate the HLA class II region spanning the HLA-DQ and -DR loci.

I found the manuscript interesting, and the authors support their investigation in their previous reports. 

I have some comments to discuss with the authors.

Please comment in the manuscript about the treatment employed in all patients, including the access among re-infected patients. Was the treatment facilitated by the study? The authors stated, "treated with azithromycin 1g" it was a single dose?

The results with HLA-DQB1 alleles are really interesting (DQB1*06 and *04), but they lack high resolution; there are several alleles in the DQB1*06 and DQB1*04 specificities, I mean alleles as DQB1*04:01 up to 04:72 and in the case of -DQB1*06:01 up to DQB1*06:99:03. It is important to include a higher resolution for both specificities identified as associated, and calculate the corresponding p-values and odds ratios.

It would be helpful to compare the alleles identified with populational data in appropriated repositories such as the Allele Frequency Net Database (AFND) and discuss similarities (or differences).

Table 4 is not particularly useful in the main manuscript; I recommend to move to the supplementary material.

Minor comments:

In Table 1, the sentence "Individuals with" is repetitive and unnecessary; please remove and indicate it in the table footnote as "subjects carrying"

In the same table, change in the variable "Age" the SD for min-max ranges.

Be sure to include the appropriate abbreviation for U.S.A., not only U.S.

Be sure to expand the abbreviations the first time of use, as in the abstract:  ...alleles with genital CT infection outcomes...

Author Response

Response to Reviewer 1 Comments

1. Summary

Thank you very much for taking the time to review this manuscript. Please find the detailed responses below and the corresponding revisions in track changes in the re-submitted files.

2. Point-by-point response to Comments and Suggestions for Authors

Comment 1: Please comment in the manuscript about the treatment employed in all patients, including the access among re-infected patients. Was the treatment facilitated by the study? The authors stated, "treated with azithromycin 1g" it was a single dose?

Response 1: Thank you for your comment. The study cohort was enrolled at a time when azithromycin 1g single dose therapy was a first line CDC recommended chlamydia treatment. All women were given azithromycin 1g single dose therapy as directly observed therapy at the time of enrollment by study staff (as part of routine care for the positive chlamydia screening test) and those who were found to be reinfected were treated again with azithromycin 1g single dose directly observed therapy by study staff. We have revised the methods to note azithromycin 1g was single dose therapy that was given as directly observed therapy in Line 389 and the same treatment was given to patients who were reinfected in Line 392.

Comment 2: The results with HLA-DQB1 alleles are really interesting (DQB1*06 and *04), but they lack high resolution; there are several alleles in the DQB1*06 and DQB1*04 specificities, I mean alleles as DQB1*04:01 up to 04:72 and in the case of -DQB1*06:01 up to DQB1*06:99:03. It is important to include a higher resolution for both specificities identified as associated, and calculate the corresponding p-values and odds ratios.

Response 2: Thank you for the comment. HLA genotyping was done to 4 digit specificity. We have updated the manuscript to provide the distribution of the DQB1*04 and *06 alleles at 4 digit specificity for the study population in Section 2.2. However, because of the sample sizes of alleles at a 4-digit specificity, it is difficult to draw any meaningful conclusions about the 4-digit allele associations. Therefore, we have also noted in Section 2.2 that 4-digit alleles were not included in the logistic regression model and that further analyses were conducted using *06 and *04 at the two-digit specificity; we also added in the Discussion section that the small cohort size limited analyses to 2-digit specificity of alleles.

Comment 3: It would be helpful to compare the alleles identified with populational data in appropriated repositories such as the Allele Frequency Net Database (AFND) and discuss similarities (or differences).

Response 3: We agree and have provided allele frequency for DQB1*04 and DQB1*06 in our cohort compared to the frequency in the Allele Frequency Net Database, noting higher allele frequencies in our cohort.

Comment 4: Table 4 is not particularly useful in the main manuscript; I recommend to move to the supplementary material.

Response 4: Thank you for your comment. However, we feel that Table 4 should be kept in the manuscript since it shows the results of the SKAT analyses, which is the basis for the association analyses that are shown in Table 5. Figure 1 shows the flow of the analyses plan and the tables we have included in the manuscript align with the flow in Figure 1.

Minor comments:

Comment 5: In Table 1, the sentence "Individuals with" is repetitive and unnecessary; please remove and indicate it in the table footnote as "subjects carrying"

Response 5: We agree with the reviewer’s comment. Suggested changes have been made in the manuscript.

Comment 6: In the same table, change in the variable "Age" the SD for min-max ranges.

Response 6: Thank you for your comment. However, since we have presented mean age, which is appropriate for analyses of this cohort, the SD is appropriate. If we had presented median age, then it would have been appropriate to provide the min-max range. Therefore, we have not revised Table 1 based on this comment, however if the editor feels we should provide the median and range in addition to mean and SD, then we can add this to the table.

Comment 7: Be sure to include the appropriate abbreviation for U.S.A., not only U.S.

Response 7: Thank you for your comment, but we believe that both U.S. and U.S.A. are commonly used abbreviations. We defer to the journal editor as to their preference for the journal and have not made any edits to this abbreviation at this time.

Comment 8: Be sure to expand the abbreviations the first time of use, as in the abstract:  ...alleles with genital CT infection outcomes...

Response 8: Thank you pointing this out. The abstract has been revised to remove the CT abbreviation for consistency with the manuscript that does not abbreviate the word chlamydia. We have used the abbreviation CT in the manuscript when specifically referring to Chlamydia trachomatis and have ensured it was spelled out initially then abbreviated as CT. Other abbreviations used in the manuscript have also been reviewed to ensure words are spelled out the first time before abbreviation.

Reviewer 2 Report

Comments and Suggestions for Authors

Author Response

1. Summary

Thank you very much for taking the time to review this manuscript. Please find the detailed responses below and the corresponding revisions in track changes in the re-submitted files.

2. Point-by-point response to Comments and Suggestions for Authors

Comment 1: Abstract:
The abstract adequately summarizes the content of the article.
Introduction:
This paragraph is too long. Please summarize it.

Response 1: We appreciate your comment, however, the introduction length for this manuscript is driven in part by important points needed to understand rationale for the study and basic methods that are done for the study so that the results can be understood by the readers (this is important because the methods section in this journal format is at the end of the manuscript after the discussion). We have made an effort to shorten the paragraphs in the introduction where possible.

Comment 2:  I would correct the sentence in line 25 “Chlamydia remains one of the most prevalent bacterial sexually transmitted infection in US and Europe”. https://www.who.int/news-room/fact-sheets/detail/sexually-transmitted-infections-(stis)

Response 2: Thank you so much for pointing it out. We have added the suggested WHO reference and cited it in the sentence on line 26 after the word “worldwide” as this is an appropriate reference to cite for worldwide summary data (that is not specific for U.S. or Europe). We have retained the reference that specifically cites U.S. data also in this same sentence. Since this is a U.S. cohort and the additional sentences in the first paragraph are referring to U.S. surveillance data, we have not added Europe in the sentence or a reference that cites chlamydia data for Europe (because we did not study a European cohort).

Comment 3:  Line 31-32: I would say something like “socioeconomic factors and recent sexual behaviors are not the only consistent predictors of CT infection” because they still have a role in
determining the risk of infection.

Response 3: We appreciate your comment. Although the sentence was accurate as stated for the cited reference, we have revised the sentence to soften the language in case our original language may have implied that behaviors/factors may not be important by saying “and there may be other factors in some individuals that contribute to the burden of chlamydia in...” in Line 33.

Comment 4: Lines 68-71 goes in methods

Response 4: We agree that this would normally go in the methods section, but since the methods section is after the discussion for this journal’s format, we think it will be important to leave this sentence in there so the readers understand this as they read the results.

Comment 5: At the end of the paragraph please add the secondary objective of the study: to find other possible genetic association sequences in HLA-DQ and -DR loci associated with CT
reinfection.

Response 5: Thank you for your comment. The study purpose that is stated in lines 73-75 already includes evaluating neighboring HLA class II variants (which includes DQ and DR) associated with reinfection risk, and thus it is not a secondary objective and that’s why we think that no further revisions are necessary.

Comment 6: Results:
The results are clear, but there are some inaccuracies:
- I suggest to report the table with the demographic characteristics of the study population

Response 6: Thank you for your comment but we feel that no further revisions need to be made in Table 1 based on this comment. As noted on line 83, this study only includes women who reported to be African American race; it did not include men or women reporting race other than African American. This is why sex at birth and race is not in the table. But we did include age in table 1.

Comment 7: Why the total population in table 1 is 166 and the study population is 185? Why you wrote that 38 participants had reinfection and in the table the total reinfected patients are 37?
Please correct or specify if you had any missing data.

Response 7: Thank you for noting this discrepancy, which we have corrected in the revised manuscript. While there were 185 women in the cohort in the previous study on HLA associations, there was only sufficient DNA samples for the next generation sequencing in 166 of the previous 185 women, and only complete data for all analysis presented in this paper for 165 women and hence, numbers were updated in table 1, 2.1, 2.3, and 4.1.

Comment 8: Discussion:
This paragraph is well written and supported by literature data.
- Why don’t you have insert in the logistic regression model the specific CD4+ IFNg and the
number of sexual intercourse? If you can’t remake the analysis, please mention it as a study
limitation.

Response 8:

Thank you for this comment. The only partner data in this cohort was number of sexual partners in last 3 months and it was not associated with reinfection risk on univariate analysis and therefore was not included in the logistic regression model. We have added a sentence at the end of section 4.4 that other non-genetic covariates were not included in the model since they were not associated with reinfection risk on univariate analysis. Regarding CT specific CD4+ IFN-g, including this immune marker in our analyses of MHC markers was not the purpose of this study. We have not performed CD4+ IFN-g testing in all the subjects who were analyzed in this study and it is a future direction of the research; we have updated the discussion to note this is a future direction.

Comment 9: I would also mention in study limitation that your population is only composed by African American women, thus limiting the generalization of your conclusion (even if the
proportion of African ancestry was low in your population).

Response 9: We agree and have added this as a limitation in the Discussion section.

Comment 10: Methods:
- I suggest to mention a paragraph “Ethical approval” with more accurate information, also
considering that it is a genetic study.

Response 10: We would like to mention that this information was already presented in the Institutional Review Board Statement section per the journal format. If the editor needs any additional information on the IRB approval, please let us know.